# Nontargeted Screening for Flavonoids in Salicornia Plant by Reversed-Phase Liquid Chromatography–Electrospray Orbitrap Data-Dependent MS^2^/MS^3^

**DOI:** 10.3390/molecules28073022

**Published:** 2023-03-28

**Authors:** Maroussia Parailloux, Simon Godin, Ryszard Lobinski

**Affiliations:** 1IPREM, UMR 5254, E2S UPPA, CNRS, Université de Pau et des Pays de l’Adour, 64000 Pau, France; simon.godin@univ-pau.fr (S.G.); ryszard.lobinski@univ-pau.fr (R.L.); 2UMR 152 PharmaDev, Université de Toulouse, IRD, UPS, 31062 Toulouse, France; 3Chair of Analytical Chemistry, Department of Chemistry, Warsaw University of Technology, 00-664 Warszawa, Poland

**Keywords:** *Salicornia*, flavonoids, metabolomics, high-resolution tandem mass spectrometry, fragment ion search, nontargeted screening

## Abstract

The Salicornia genus has great potential in agrifood industries because of its nutritional benefits related to its high content of antioxidant compounds, including flavonoids. A nontargeted method based on reversed-phase liquid chromatography–electrospray orbitrap data-dependent MS^2^/MS^3^ and the fragment ion search (FISh) strategy was developed to screen flavonoids in Salicornia plants. An extensive study of fragmentation of a set of flavonoid standards allowed for the definition of 15 characteristic fragment ions for flagging flavonoids in the plant matrix. The nontargeted analysis was applied to *Salicornia europaea* species and allowed for the annotation of 25 candidate flavonoids, including 14 that had not been reported previously. Structural prediction of two unreported flavonoids and their isomeric forms was based on an advanced data processing method using an in silico approach and in-house databases compiling flavonoid-specific chemical substitution. Finally, the method developed allowed for the optimization of extraction yields of flavonoids from the plant matrix.

## 1. Introduction

Currently, there is growing interest in the discovery and the promotion of marine bioactive compounds. Most scientific research efforts have been concentrated on the mining of algal-derived components [1,2], while the biochemical value of marine coastal plants, such as the Salicornia genus, has not been largely explored yet [3,4,5].

Salicornia, also known as sea bean or samphire, is a halophytic plant growing on salt-saturated areas [6,7,8,9] and classified into the *Caryophyllales* plant order, which regroups more than 30 species reported in the Plants of the World Online and eHaloph databases [4]. Salicornia has been domesticated [5,10,11,12,13,14,15], and is traditionally used in the human diet for its nutritional benefits due to its high content of natural minerals, dietary fibers, polysaccharides, phytosterols, phenolic acids and flavonoids [16,17,18].

Flavonoids have drawn particular attention because of their numerous biological activities as, e.g., antioxidants, UV quenchers, antitumoral agents, anti-inflammatories, cardiovascular protectors and antidiabetic and antiobesity agents [16,19,20,21]. Flavonoids consist of ten main classes produced from the phenylpropanoid pathway, starting with chalcone precursors, and show two common benzene rings (A and B) linked to a heterocyclic pyrane ring (C) usually displaying a ketone group (C4), a hydroxyl group (C3) and/or a π bond (C2–C3) (Figure 1) [8,22,23].

The differentiation of flavonoids within a same class is based on the nature and the position of substituents on the A and B rings [24,25]. The C ring constitutes an α-pyrone associated to the benzene ring (A) and linked to B ring either on C2 (flavones) or C3 (isoflavones). Methoxy-, C-3/C-7 sulfated, 3-O/7-O and C-6/C-7 glycosylated forms of flavonoids have been also reported in the literature [20,21].

Since the 2010s, more than twenty flavonoids have been identified in the Salicornia genus, principally involving isoflavones, flavanones, flavonols, flavonol glycosides, e.g., isorhamnetin-3-β-d-glucoside or quercetin-3-β-d-glucoside and some anthocyanin derivatives [26,27,28]. Most of the techniques used for characterization are based on high-performance liquid chromatography (HPLC) with diode array detection (DAD). This method allows for the monitoring of possible shifts, due to the presence of hydroxyl groups and their substitutions, in absorption wavelengths of the two maxima: λ_max_—320–385 nm (band I, B ring); and λ_max_—250–285 nm (band II, A ring) [14,15,29,30,31]. However, because of the lack of sensitivity of spectrophotometric detection and the likely overlapping of the λ_max_ of flavonoids [32], alternative methods based on nuclear magnetic resonance (NMR) spectroscopy [21] or tandem mass spectrometry (MS/MS) [27,29] analysis were proposed for their structural elucidation.

Studies using 2H [21,30] and 13C [30] NMR analysis permitted the identification of flavonoids and some of their glycosylated derivatives, including the novel compound isoquercitrin-6″-*O*-methyloxalate in *Salicornia herbacea* L. [21]. Although NMR analysis allows for the identification of previously unreported compounds, it requires time-consuming steps of purification and concentration to obtain measurable signals [32]. The formula prediction of flavonoids in complex mixtures can be achieved with high-resolution accurate mass (HRAM) MS analysis on the basis of their accurate mass, isotopic spacing and ratio [27,30]. Nonetheless, a straightforward identification only can be achieved by the use of MS/MS fragmentations [29,30,31]. However, these current mass spectrometry approaches require purified standards, which are hardly available while the reference spectra present in the data refer to the previously identified compounds only, and are unable to cover the structural diversity of flavonoids [29].

The emergence of a structure-based fragment triggering method offers perspectives for compounds’ annotation and discovery by means of their fragmentation patterns [33,34,35,36]. The objective of this work was to develop a nontargeted HRAM MS^n^ approach for a comprehensive flavonoid profiling in a marine plant, *Salicornia europaea*, based on the definition of flavonoid class-specific fragment ions and neutral losses.

## 2. Results and Discussion

### 2.1. Method Development

#### 2.1.1. Fragmentation Pathways of Flavonoids

The purpose was to determine a set of class-specific fragment ions by the targeted MS^2^ fragmentation of gallic acid and 14 flavonoid standards representing five different flavonoid subclasses. Different collision modes and energies were tested in order to select the fragmentation conditions to generate fragment ions common to every of them. The monitoring of the flavonoid fragmentation pathways, the annotation of their characteristic fragment ions and their related neutral losses are shown in Table 1.

The fragment ion distribution in the MS^2^ data obtained for every flavonoid standard was compared and revealed two common fragment ion populations. The first was generated at low fragmentation conditions (HCD30, 40 and 50) and involved eight candidate fragment ions with *m*/*z* values higher than or equal to *m*/*z* 199 (*m*/*z* 199.0401, *m*/*z* 211.0401, *m*/*z* 225.0556, *m*/*z* 227.0350, *m*/*z* 243.0299, *m*/*z* 255.0299, *m*/*z* 271.0248 and *m*/*z* 285.0405, respectively). The signal intensity of most of these fragment ions tended to decrease drastically by the increase in the fragmentation collision energy, and completely disappeared in the HCD80 spectra. These fragment ions were usually generated through decarboxylation and other mechanisms releasing, for instance, CH_4_, CH_2_O, C_2_H_2_O and C_3_H_6_O_2_ losses. For glycosylated flavonoids, they came from the [(M-H)-glycoside] moiety, as it was observed in their HCD30, 40 and CID30 spectral data (Appendix A). In the present panel of flavonoid standards, the neutral loss of C_6_H_10_O_5_, corresponding to the glucose moiety (162.0523 Da), attested to the presence of O-glycosylated structures [39]. Sugar loss, corresponding to rutinose (C_12_H_20_O_9_), was also monitored in the fragmentation pathway of hesperidin.

The second Ion population was obtained in higher fragmentation conditions. A set of seven flavonoid-specific fragment ions were identified in a mass range lower than *m*/*z* 170, especially starting from HCD60. Globally, they appeared in HCD60 conditions and reached their highest abundance at the HCD80 level. The fragment ions *m*/*z* 83.01385, *m*/*z* 107.0139, *m*/*z* 125.0244 and *m*/*z* 151.0037 were found in all mass spectral data of the flavonoid standards. The fragment ions *m*/*z* 123.0452 and *m*/*z* 133.0295 were also selected due to their high abundance in HCD60 and HCD80 MS^2^ scans in spectral datasets of nine flavonoid standards. The appearance of *m*/*z* 169.0142 and *m*/*z* 285.0405 ions was noticed in HCD60 and HCD80 MS^2^ scans of three flavonoid precursors and matched the exact mass of gallic acid, kaempferol and luteolin.

It is worth mentioning that fragmentation data acquired at HCD60 were useful for the differentiation of flavonoid isomers by their individual infusion. Indeed, different fragmentation tendencies were observed for the isomeric couple kaempferol/luteolin and apigenin/genistein. In HCD60 conditions, it was noticed that kaempferol showed an array of abundant fragment ions below *m*/*z* 150, whereas luteolin displayed a predominant fragment ion, which was the *m*/*z* 133.0291 (Figure 2a). Similar fragmentation behavior was observed for the apigenin, which displayed the high intense fragment ion *m*/*z* 117.0316, whereas genistein was recognized by its intense fragment ion *m*/*z* 133.0291 (Figure 2b).

The stepped fragmentation mode was tested to compile both fragment ion sets detected in HCD40 and HCD80 spectra and cover all the flavonoid standards differing in subclass, methylation and glycosylation substitution. Fixing the fragmentation conditions at HCD60 ± 20 was the best compromise to simultaneously produce all of the 15 characteristic fragment ions appearing in the *m*/*z* 80–200 range.

The targeted fragmentation of infused flavonoid standards allowed for the selection of a set of 15 characteristic fragment ions. Flavonoid-specific fragment ions were determined on the basis of their capability to flag any flavonoid standard independently to their subclass, glycosylation and methylation level. Stepped collision energies fixed at HCD60 ± 20 allowed for the regrouping of milder fragmentation data generated in HCD40 and the monitoring of Retro-Diels–Alder mechanisms occurring on the flavonoid C-ring in drastic fragmentation conditions (HCD80). In such a strategy, it was possible to mine flavonoids with different fragmentation behaviors by the detection of all the characteristic fragment ions and an optimization of their signal intensities. For instance, glycosylated flavonoids produced intense characteristic fragment ions higher than *m*/*z* 200 related to the decomposition of their [(M-H)-glycoside] moiety, while aglycone forms generated intense characteristic fragment ions lower than *m*/*z* 199 under the same fragmentation conditions. As a result, stepped collision mode was found to be a viable strategy to cover all the typical fragmentation pathways in a wide mass range, as was recently proposed for glycosylated flavonoids by Cerrato et al. [35].

Most of the fragment ions selected to screen flavonoids originated from the opening of their common C-ring, related to the Retro-Diels–Alder (RDA) and retrocyclization mechanisms. Indeed, the fragment ions *m*/*z* 107.0139, *m*/*z* 125.0244 and *m*/*z* 151.0037 have been already identified in the literature for a dozen model flavonoids, which confirmed their specificity to this compound class [34,40,41,42]. Bigger fragment ions, including *m*/*z* 225.0556, *m*/*z* 227.0350 and *m*/*z* 243.0299 have been also reported, especially in fragmentation data of flavone and flavonol subclasses [37]. Notably, the smallest fragment ion of the set, *m*/*z* 83.01385, was detected here for the first time and selected as a characteristic fragment ion because of its ubiquity in all flavonoid spectra. The study and the comparison of the fragmentation pathways of gallic acid with the other 14 flavonoid standards assumed that the fragment ions at *m*/*z* 169.0142, *m*/*z* 125.0244 and *m*/*z* 83.0139 can be also used to flag phenolic acids usually considered as flavonoid derivatives because of their involvement in the synthesis of some flavanols and gallotannins in the phenylpropanoid pathway [22,35].

Note that the infusion of flavonoids has been used as a time-saving strategy allowing for the optimization of the collision energies tested and the monitoring of the behavior of standards at controlled voltage conditions independently to their retention times over a classical chromatographic separation. Additionally, fragmentation studies in CID30 and HCD30 conditions can be complementarily used to confirm structural relationships between the fragment ions from the first fragmentation steps of precursors and the detection of glycosylated and polymeric forms of flavonoids generally abundant in plant extracts [42].

In this way, the targeted fragmentation of infused flavonoid standards allowed for an untargeted Top3 OT ddMS^2^/MS^3^ method using HCD60 ± 20 mode to produce a set of 15 characteristic fragment ions for the annotation of flavonoids in *Salicornia*.

#### 2.1.2. Validation of the Fragment Ion Set for the Flavonoid Annotation

In order to validate the capability of the selected fragment ion set to cover efficiently the structural diversity of flavonoids, the untargeted Top3 ddMS^2^/MS^3^ method was applied on the 15 standards separated by reversed-phase UPLC. Chromatographic peaks of the flavonoid standards were acquired and matched with the extracted ion chromatogram (XIC) HCD60 ± 20 MS^2^ traces compiling all the 15 characteristic fragment ions, as shown in Figure 3.

As was expected, XIC MS^2^ traces of the 15 fragment ions covered all the retention times of the flavonoid standards. Moreover, the superposition of the XIC MS^2^ traces with the chromatographic separation revealed the distribution and the relative abundance of fragment ions in HCD60 ± 20 MS^2^ OT scans acquired for the 15 flavonoid standards. Of note, XIC MS^2^ traces for fragment ions higher than *m*/*z* 200 coincided with retention times of glycosylated flavonoid standards such as quercetin-3-β-d-glucoside, apigenin-7-β-d-glucoside, hesperidin and rutin. On the contrary, retention times of aglycone flavonoids corresponded to the XIC MS^2^ trace of fragment ions lower than *m*/*z* 199.

It is noteworthy to mention that such an untargeted method based on the fragment ion search strategy permits a large inventory of class-specific compounds in spite of their variable intensities and the complexity of the plant extract matrix. The developed untargeted mass spectrometry method and data processing approach were thus proposed to treat speedily metabolomic big data without requiring either preliminary time-consuming extraction procedures or fractionation treatments to simplify the sample matrix.

These observations confirmed that the set of fragment ions selected was suitable for the inventory of the flavonoid compound class independently to their structural variability and their subsequent fragmentation behavior.

#### 2.1.3. Method Validation: Nontargeted Analysis of a Model Sample

To confirm the applicability of the developed method in complex plant matrices, a grape seed powder well known for its richness in flavonoids was selected as a model sample and prepared in 10 mM NH_4_Ac pH 5.4.

Data-processing performed on Compound Discoverer^TM^ software to mine candidate flavonoids in the model sample allowed for the detection of 1068 compounds. Application of filtering criteria from the data processing workflow permitted the selection of 27 final exact masses. Because of their wide structural diversity and their subsequent fragmentation pathways, a threshold of a minimum of three class-specific fragment ions observed in their HCD60 ± 20 MS^2^ OT scans was used to annotate putative flavonoids. All the candidate flavonoids were detected with a mass error lower than 5 ppm and an area peak max greater than or equal to 1 × 10^4^. The use of *Search Mass Lists* node permitted the identification of 20 compounds based on their exact mass and their match with online and local databases. Notably, eight of them showed multiple matches with flavonoid isomers (Figure 4).

In the datasets, 12 flavonoids and 6 phenolic acids were found in grape seed extract. Most of them were flavan-3-ols, flavonols and phenolic acid derivatives detected in a glycosylated form, such as catechin-hexoside. Moreover, three exact masses matched with type-B procyanidins. Notably, the nontargeted method permitted the detection of seven unreported flavonoids with their unidentified isomers in the model sample by the detection of characteristic fragment ions in their HCD60 ± 20 spectra. These observations thus confirmed the efficiency of the nontargeted ddMS^2^/MS^3^ method to tag candidate flavonoids and phenolic acids structurally related to gallic acid in complex matrices by the use of the selected set of 15 characteristic fragment ions.

Additional CID30 MS^2^/MS^3^ scans were triggered for candidate flavonoids showing class-specific fragment ions in their HCD60 MS^2^ scans in order to detect possible glycoside loss. For precursor ions higher than *m*/*z* 600, CID30 MS^3^ scans were triggered on the biggest fragment ions detected in HCD60 ± 20 MS^2^ scans, thus revealing the structural relationships between the characteristic fragment ions used for the screening of flavonoids. For instance, CID30 MS^3^ fragmentation data of the unknown flavonoid *m*/*z* 865.1964 were acquired for the fragment ions *m*/*z* 255.0927, *m*/*z* 243.0296 and *m*/*z* 227.0346, and in return generated other characteristic fragment ions, e.g., *m*/*z* 199.0401 and *m*/*z* 211.0701 found in prior HCD60 ± 20 (Appendix A).

Mass spectral data acquired with the model grape seed sample allowed for confirmation of the efficiency of the untargeted method to mine and annotate flavonoids in complex plant matrices by the use of 15 characteristic fragment ions. Although CID30 MS^2^/MS^3^ data acquired in the ion trap did not permit an unambiguous structural elucidation due to the lack of resolution, they can be used to confirm the positive detection of candidate flavonoids showing characteristic fragment ions in their prior to HCD60 ± 60 MS^2^ scans. Nonetheless, fragmentation data acquired in HCD60 ± 20 provided structural information in agreement with the literature for the identification of detected flavonoids, as, for instance, polymeric forms of catechins generating procyanidin derivatives, or glycosylated forms of flavonoids by the observation of their intense [(M-H)-glycoside] moiety [43,44].

### 2.2. Method Application

#### 2.2.1. Method Application for the Nonscreening of Flavonoids in *Salicornia europaea* Extracts

The developed nontargeted Top3 ddMS^2^/MS^3^ method was carried out to mine flavonoids in *Salicornia* extracts prepared either with 30 mM Tris-HCl pH 7 or 10 mM NH_4_Ac. Data processing performed with a nontargeted workflow designed using Compound Discoverer^TM^ software allowed for the detection, in total, of 25 candidate flavonoids after applying filtering criteria (background is false; −5 ≤ mass error ≤ 5 ppm; area peak max ≥ 1 × 10^4^; class-specific fragment ions ≥ 4). An additional filtering option was used to select only candidate flavonoids showing CID30 MS^2^/MS^3^ scans triggered after the detection of characteristic fragment ions (MS depth less than or equal to 3). All the detected compounds were reported with a mass error lower than 5 ppm, as shown in Table 2.

In the dataset, the use of the node *Search Mass Lists* permitted the fast identification of candidate flavonoids: five glycosylated flavonoids were detected and matched with flavonoids reported in online and local databases. The identification of quercetin-3-β-d-*O*-glucoside, quercetin-3-(6″-malonyl)-glucoside, isorhamnetin-hexoside, luteolin and its di-glycosylated form confirmed their specificity to the *Salicornia* genus [21,26,30,31]. These observations are in accordance with the literature, which reports quercetin-glycoside derivatives as typical flavonoid structures in Salicornia [45]. Flavonoid glycosides were easily elucidated by the detection of their [(M-H)-glycoside] moiety and their related neutral loss, revealing the basic structure of the glycoside group, e.g., hexose or pentose. Likewise, the structural annotation of quercetin-3-(6″-malonyl)-glucoside can be easily identified by its characteristic malonyl-hexose loss [39,42,45]. Nonetheless, the fragmentation data did not permit the precise definition of the position of the glycoside group on the flavonoid skeleton core and their exact nature, as was reported for isorhamnetin-hexoside and luteolin-di-hexoside. However, the elucidation of the glucose and its position in the structure of quercetin-3-β-d-*O*-glucoside was confirmed, with its similar retention time and fragmentation data to the available standard.

Note that 4′-OH-5,7,2′-trimethoxyflavanone 4′-rhamnoside and luteolin-*O*-sulfate are reported here for the first time in the *Salicornia europaea* sample. Moreover, three flavonoids displayed multiple matches: more than 30 structural propositions were given from local databases, e.g., *Arita Lab 6549 Flavonoid Structure Database,* and corresponded to mono- and polymethoxylated flavonoids classified mostly in flavones, isoflavones and flavonols subgroups.

Interestingly, 14 candidate flavonoids were not reported in the literature because no match was observed with local databases. Nevertheless, the observation of the *m*/*z* 301.0354 fragment ion in mass spectral data of *m*/*z* 505.0983 assumed its structural affiliation with a quercetin-acetyl glucoside, which may be quercetin-3-*O*-(6″acetyl glucoside). Indeed, Alves et al. recently demonstrated that the fragmentation of quercetin-3-(6″-malonyl)-glucoside leads to the decarboxylation of its malonyl substitution and the appearance of the product ion matching unambiguously with the structure of *m*/*z* 505.0983, displaying an acetyl glucose group [45]. Similarly, the unknown compound *m*/*z* 519.1144 showed the fragment ions *m*/*z* 315.0505 and *m*/*z* 151.0039, related to the isorhamnetin and its Retro-Diels–Alder rearrangement (Figure 5a,b). Data processing also revealed three other sulfated flavonoids additionally to the luteolin-*O*-sulfate (*m*/*z* 364.9973) by the detected loss of sulfonate (79.957 Da) and the appearance of the intense characteristic fragmentation pathway of luteolin ([M-H]^−^
*m*/*z* 285.0399) with *m*/*z* 133.0291 (^1,3^B^−^ moiety), as was found for its di-glycosylated form (Figure 5c).

In summary, the developed nontargeted analysis based on a set of 15 flavonoid-specific fragment ions included in a Top3 ddMS^2^/MS^3^ method enabled the detection of previously unreported flavonoids in *Salicornia europaea*plants. Data processing based on the annotation of flavonoid-specific fragment ions in HCD60 ± 20 spectra offered an overview on the presence of flavonoids in a wide mass range *m*/*z* [280–1120 Da] and their possible affiliations to different subclasses. Most of the flavonoids identified in *Salicornia europaea* extracts were quercetin, luteolin and isorhamnetin derivatives [21,26,30,31]. For the first time, four sulfated flavonoid derivatives were detected In *Salicornia europaea*.

If this method is relevant to cover all the candidate flavonoids in *Salicornia* extract, it is limited in terms of distinction of isomeric forms. Although fragmentation data based on Retro-Diels–Alder mechanisms and retrocyclization could provide information in HCD60 ± 20 MS^2^ scans on the number of flavonoid substituents and their possible location on the A, B and C rings, they did not permit the positioning of them on a specific carbon of their skeleton ring. Moreover, the abundance of isomeric forms renders the manual structural elucidation of candidate flavonoids fastidious and inaccurate. Structural characterization of flavonoids can be performed with NMR analysis. However, these methods remain time-consuming and require a rigorous separation of detected compounds [32]. To cope with this limitation of the method, an innovative data processing method using a structural transformation database driven workflow on Compound Discoverer^TM^ software was designed for the characterization of unknown candidate flavonoids and distinction of their isomeric forms.

#### 2.2.2. Identification of Flavonoids in *Salicornia europaea* Extracts Based on In Silico Predictive Combination Flavonoid-Specific Chemical Substitutions in Known Parent Molecules

Flavonoids for which the structural identity was not assigned or resulted in multiple matches were identified by a data processing method assisted with a list of predictive combinations of flavonoid-specific chemical substitutions. For this purpose, the unknown candidate flavonoids were characterized based on the structures of flavonoids reported in previous studies on the *Salicornia* genus and *Arita* databases. The calculation of their mass shifts and the prediction of the most likely structural prediction are reported on Table 3. The calculation of the FISh score was given to strengthen confidence in their identification.

Chemical substitutions were predicted with a mass error of less than 3 ppm and a FISh score greater than 30% for nine candidate flavonoids. Structures were predicted on the basis of derivatives of luteolin and quercetin-3-β-d-glucoside, naringenin, genistein, hesperetin and gallic acid. The most frequently observed structural predictions were O-methylation, acylation and methyloxalate addition. As an example, the acylation transformation applied in silico on the candidate structure of quercetin-3-β-d-glucoside permitted the structural identification of the unelucidated structure of *m*/*z* 505.0987, corresponding to the quercetin-3-*O*-(6″acetyl glucoside) [42,45].

On the basis of the structural prediction by the software, examples of hypothetical structures were proposed for the characterization of six candidate flavonoids, affiliated to flavanone, flavone and flavonol subclasses, as is shown in Table 4.

The structural predictions attributed to the parent compounds revealed O-methylated flavonoids, acetylated flavonoids and flavonoid methyloxalate-glycosides, in accordance with the literature [21]. It must be noted that the structures proposed were solely elucidated by means of the basic core of parent compounds, and the position of arriving groups was not definitively fixed because of the possible occurrence of isomers. In spite of the correlation between the predicted structure and the experimental fragmentation data, the discrimination between isomeric couples can be ambiguous, especially when the fragmentation pathways did not affect directly substituted groups. Nonetheless, an individual calculation of the FISh score can be performed for all the possible structures to select the most reliable one regarding the experimental fragmentation data acquired.

Structural prediction guided by a flavonoid-specific chemical substitution list allowed for the attribution of putative structures of nine novel candidate flavonoids. Although the developed data processing method did not systematically give a single structural prediction for flavonoid isomers, it permitted the simplification of the characterization of unknown compounds by delivering information on their affiliations to different compound subclasses and their likely structural relationships with parent compounds.

#### 2.2.3. Evaluation of the Extraction Efficiency of Flavonoids from Salicornia Plant

Due to its antioxidant virtues and its nutritional benefits, Salicornia has been the subject of recent studies dealing with quantitative analysis or semiquantitative estimation of polyphenols in its dry matter, including phenolic acids and flavonoids. Usually, the global proportion of flavonoids is expressed according to the amount or the extraction percentage of quercetin-3-β-d-glucoside, for which its standard remains easily available [46,47].

To compare the global distribution of flavonoids and the individual proportion of each candidate compound in the total fraction, the developed nontargeted method was applied to the *Salicornia e europaea* extracts prepared in triplicate using either 30 mM Tris-HCl pH 7 or 10 mM NH_4_Ac pH 5.4 under sonication. Extractions were performed in acidic media to improve the release of flavonoids present in forms of esters, glycosides or polymers in plant matrices and to prevent their oxidative degradation over their extraction assisted with ultrasounds or their further long-term storage [46,48,49].

Because of the lack of extended flavonoid standards to quantify the newly detected candidate compounds, the extraction performances were evaluated by a comparison of the number of candidate compounds and their area peaks in both extraction conditions.

A comparative study based on the total sum of area peaks of candidate flavonoids in both extracts demonstrated a significant difference with a higher flavonoid recovery for the extraction procedure based on sonication 1.7 × 10^6^ ± 2.5 × 10^5^ in Tris-HCl and 2.8 × 10^7^ ± 5.8 × 10^6^ in NH_4_Ac extracts, respectively. Nonetheless, an individual study of area peaks acquired for every flavonoid showed that 12 of them were extracted more efficiently in NH_4_Ac under sonication, whereas 5 others were preferentially extracted in Tris-HCl, as is shown in Figure 6.

Indeed, their single annotation in HCl or NH_4_Ac extracts assumed that their area peaks was in one of the extraction conditions below the threshold fixed at 1 × 10^4^, according to the result-filtering criteria applied with Compound Discoverer^TM^ software. Note that *m*/*z* 505.0989, *m*/*z* 519.1144 and *m*/*z* 387.1085, presuming to be quercetin-3-*O*-(6″acetyl glucoside), isorhamnetin-7-*O*-(6″acetyl glucoside) and 7-acetyl-5-hydroxy-3′,4′,5′ trimethoxyflavanone, respectively, displayed the highest signal intensities, especially in the NH_4_Ac extract.

Quantitative analysis of quercetin-3-β-d-glucoside and luteolin was carried out with the standard addition method in *Salicornia* extracts prepared in 30 mM HCl pH 7 or 10 mM NH_4_Ac pH 5.4. The final concentrations of quercetin-3-β-d-glucoside and luteolin were determined at 3.85 µg.mL^−1^ ± 0.09 and 0.87 µg.mL^−1^ ± 0.02 in 30 mM HCl and 3.15 µg.mL^−1^ ± 0.10 and 1.83 µg.mL^−1^ ± 0.07 in 10 mM NH_4_Ac.

The Id nontargeted method permitted the estimation of flavonoid recovery in both extracts and optimization of their extraction efficiency. Ultrasound-assisted extraction in NH_4_Ac improved the extraction efficiency in terms of the quantity and the number of candidate flavonoids. Indeed, the use of ultrasounds may help disrupt cell walls, favoring the release of flavonoids interacting with cellular matrix [22,32]. Moreover, the application of a lower pH in NH_4_Ac medium may increase the extraction efficiency, as it was reported in the literature [48,49]. In both extraction conditions, high amounts of quercetin-3-β-d-glucoside and luteolin were achieved in *Salicornia europaea* and may confer valuable antioxidant properties [20,45,50]. Moreover, the high concentrations of the characterized new structure of 6-methoxy-luteolin-sulfate-*O*-(dimethyloxalate-glucoside), following by quercetin-3-*O*-(6″acetyl glucoside) and isorhamnetin-7-*O*-(6″acetyl glucoside), suggest the preferentially synthesis of sulfated and acetylated forms of quercetin and isorhamnetin in *Salicornia europaea*.

## 3. Materials and Methods

### 3.1. Biological Materials, Chemical Reagents

Myricetin, apigenin, genistein, naringenin, catechin, baicalin and hesperetin were purchased from Merck Sigma-Aldrich (Fontenay-sous-Bois, 94120, France). Gallic acid, quercetin, rutin, hesperidin, luteolin, kaempferol, quercetin-3-*O*-β-d-glucoside and apigenin-7-glucoside were purchased from LGC Standards (Molsheim, 67120, France). Stock solutions of 500 µg.ml^−1^ were prepared for every flavonoid standard in pure MeOH and stored at −20 °C. Samples of *Salicornia europaea* were obtained from Terre Saline (Charente-Maritime, 1700, France). A sample of grape seed powder (*Vitis vinifera*) was obtained from Joli’essence (La Roque d’Anthéron, 13640, France) and used as model sample for method development. *Salicornia europaea* samples were carefully washed three times with water, freeze-dried and ground into powder.

Methanol and acetonitrile were LC-MS-grade and purchased from Honeywell (Morris Plains, NJ, USA). Ammonium acetate, acetic acid and deuterium were LC-MS-grade and were purchased from Sigma Aldrich (L’Isle D’Abeau Chesnes, 38080, France). Ultrapure water was obtained from a Direct-Q 3 UV (Merck, Fontenay-sous-Bois, 94120, France).

### 3.2. Sample Preparation

Flavonoids were extracted in *Salicornia europaea* and grape seed powders by the use of two different techniques: extraction with 30 mM Tris-HCl (pH 7.3) and ultrasound-assisted extraction in 10 mM NH_4_Ac (pH 5.4). Both extraction techniques were performed at a solid-to-liquid ratio of 1:20. Extraction with 30 mM Tris-HCl was carried out for 2 h under shaking at room temperature. Extraction with 10 mM NH_4_Ac was performed in an ultrasonic bath for 1 h with an ultrasound frequency of 42 kHz and ultrasonic power of 100 W at room temperature. Crude extracts were then ultracentrifuged over 20 min at 50,000 rpm at 21 °C. Supernatants were collected and stored at −20 °C. Exposure to direct sunlight was avoided during the sample preparation.

### 3.3. Instrumentation

Analysis of flavonoids was carried out using an Ultimate 3000 RSLC system (ThermoFisher Scientific, Bremen, Germany) coupled with an Orbitrap Fusion Lumos Tribrid mass spectrometer (ThermoFisher Scientific, Waltham, MA, USA) operated in negative mode. Data processing for the structurally intelligent annotation of both known and unreported structures was carried out using Compound Discoverer 3.2^TM^ (ThermoFisher Scientific, Waltham, MA, USA). In parallel, Mass Frontier 7.0^TM^ (HighChem, Bratislava, Slovakia) was used for the structural prediction of novel flavonoids and their in silico fragmentation.

### 3.4. Chromatographic Separation

The separation of flavonoids was performed on an Acquity UPLC BEH C18 column (150 × 2.1 mm, 1.7 µm, 130 Å) (Waters, Saint-Quentin-en-Yvelines, 78180, France) at 40 °C. The mobile phases were 0.1% formic acid in water (A) and 0.1% formic acid in acetonitrile (B). HPLC separation was carried out at a flow rate fixed at 0.4 mL.min^−1^ with the following gradient elution profile: 0–2 min, 10% B; 2–10 min, 10 to 90% B; 10–12 min, 90% B; 12–13 min, 90 to 10% B; and 13–15 min, 10% B. An aliquot of 15 µL of diluted extract was injected.

### 3.5. Untargeted Screening of Flavonoids Based on Fragment Ion Search (FISh) Strategy

The determination of flavonoid-specific fragment ions for their screening in *Salicornia europaea* extracts was performed with a direct-infusion tandem mass spectrometry analysis. A set of flavonoid standards were infused at 100 ng.ml^−1^ at the flow rate 5 µL.min^−1^. Collision-induced dissociation (CID) and higher-energy C-trap dissociation (HCD) fragmentation modes were tested at different collision energies (30, 40, 60 and 80). The CID parameters were defined with an activation time of 10 ms and activation Q at 0.25 (the Q parameter determines the stability of the precursor ion’s trajectory in the ion trap and is directly linked to the RF voltage amplitude; it defines, together with fragmentation energy, the depth of fragmentation obtained and the value of the low-mass cutoff). The ESI parameters were sheath gas at 5 (a.u.), auxiliary gas at 0 (a.u.), sweep gas at 0 (a.u.) and ion transfer tube temperature at 275 °C.

The flavonoid contents in *Salicornia* and grape seed extracts were analyzed by means of LC-MS with a Top3 data-dependent MS^2^/MS^3^ (ddMS^2^/MS^3^) acquisition mode (Figure 7).

Electrospray ionization conditions were sheath gas 50 (arb), auxiliary gas 10 (arb), sweep gas 1 (arb), spray voltage 2500 V in negative mode and rf lens 50%. Ion transfer tube and vaporizer temperatures were fixed at 350 °C. For the top3 ddMS^2^/MS^3^ analysis, scan events are detailed on Figure 8.

Full MS scans were acquired in orbitrap (OT) at resolution 120,000, scan range *m*/*z* 150–1500, AGC target: 60%, maximum injection time: 250 ms, dynamic exclusion 5 s and intensity threshold 2 × 10^4^. To produce flavonoid-specific fragment ions, ddMS^2^ OT scans were used. Stepped fragmentation, which consists of applying several fragmentation energies for a single MS2 spectrum, was more prone to give satisfying fragmentation in this case, and was thus used. Parameters for these ddMS2 OT scans were resolution 60,000, stepped fragmentation conditions HCD60 ± 20, isolation window width 1.6 Da. For the HCD60 ± 20 MS^2^ event, two scan ranges and lists of flavonoid-specific fragment ions were defined: HCD60 ± 20 MS^2^ with a scan range *m*/*z* 150–600 was dedicated to detect aglycone flavonoids and produce characteristic fragment ions between *m*/*z* 80–170 to trigger further CID30 ddMS^2^ ion-trap (IT) scans; in parallel, HCD60 ± 20 MS^2^ was performed to fragment polymeric and glycosylated forms ranging from *m*/*z* 600–1500 and produce characteristic fragment ions between *m*/*z* 150–272 to trigger CID30 ddMS^2^ IT scans. CID30 ddMS^2^/MS^3^ scans were performed to confirm the detection of flavonoids and displayed the following settings: scan rate 33,333 Da/s, peak width 0.5 FWHM, isolation width 2 Da. CID30 ddMS^3^ IT scans were applied on the highest *m*/*z* values of fragment ions detected in HCD60 ± 20 by the use of mass trigger settings.

For every plant extract obtained in 30 mM Tris-HCl or 10 mM NH_4_Ac buffers, raw data obtained in Top3 ddMS^2^/MS^3^ OT acquisition were processed on Compound Discoverer 3.2^TM^ software by the use of an untargeted workflow (Figure 9).

The flagging of candidate flavonoids was based on a computational labeling of characteristic fragment ions determined in a prior-targeted fragmentation of the flavonoid standards. All the compounds showing fragmentation patterns specific to the flavonoid compound class were listed in a final table of results with a mass tolerance lower than 5 ppm and a minimum peak intensity at 1 × 10^4^. For this purpose, the untargeted workflow was based on the selection of spectra in raw data and alignment of retention times between samples, with a maximum shift of 0.2 min and a minimum signal/noise ratio at 20. The *Detected Compounds* node was used to detect all the precursor ions showing at least the molecular formula C H O and a maximum element threshold with C_90_, H_190_, O_50_ and S_10_, assuming the possible detection of glycosylated, polymeric and sulfated flavonoids. The *Compound Class Scoring* node calculates the number of characteristic fragment ions found in HCD60 ± 20 MS^2^ scans of detected candidate flavonoids. A minimum of four characteristic fragment ions were used to consider a positive flavonoid flagging. The *Search Neutral Loss* node was used to screen characteristic neutral and radical losses from the fragmentation pathways of the flavonoid compound class. The *Create Mass Trace* node allowed for the plotting of XIC traces based on the retention times of flavonoids for which characteristic fragment ions were detected in HCD60 ± 20 MS^2^ OT scans. Additional parameters such as *Fill Gaps* or *Mark Background* were used to indicate missing peaks, correct alignment errors and discriminate peaks from blanks. Finally, the node *Search Mass List* was applied in order to annotate compounds using local databases, i.e., an in-house-created database containing 85 flavonoids, and the software included Arita Lab 6549 Flavonoid Structure Database. In parallel, the *Search mzCloud* and *Search ChemSpider* nodes were used to interrogate online molecular and spectral databases. Filtering criteria were applied to select candidate flavonoids detected in plant extracts with the following settings: background is false; −5 ≤ mass error ≤ 5 ppm; area peak max ≥ 1 × 10^4^; class-specific fragment ions ≥ 2; MS^2^ equal to data-dependent for preferred ion; MS depth less than or equal to 3.

### 3.6. Untargeted Screening of Flavonoids Based on Fragment Ion Search (FISh) Strategy

Structural prediction of unknown flavonoids and isomeric forms was performed with a data processing workflow combining an in-house-created exhaustive list of chemical functions and substitutions previously reported in flavonoids, called transformations, with structures in the local in-house database of 85 flavonoids, serving as parent molecules for the transformations. The software virtually creates a list of formulas resulting from all possible combinations of transformations with a parent molecule, and then attempts to find matching signals for these formulas. The reporting of an identified compound is then conditioned by the presence of fragments matching the original fragments of the parent molecule in the associated MS2 spectrum, as well as fragments showing a mass shift matching the mass of the transformation (or the sum of them if several transformations are involved). The location of the substitution on the structure of the parent molecule was then attempted based on which fragments would show the mass shift and which were unmodified compared the corresponding known flavonoid. For every structural prediction, a Fish score was given with an *m*/*z* value matching within 5 ppm to one a of the predicted flavonoid structure (Figure 10).

### 3.7. Evaluation of Extraction Performances of Flavonoids in Salicornia Europaea Extracts

Extraction recovery was evaluated with the area peak of candidate flavonoids detected in both extraction conditions (30 mM Tris-HCl and 10 mM NH_4_Ac, respectively). Salicornia extracts prepared in triplicate in both extraction conditions were analyzed by nontargeted ddMS^2^/MS^3^. Statistical analysis using *t*-test was applied to confirm significant differences between both methods. In all cases, differences were considered statistically significant at *p* < 0.05. Flavonoids for which the flavonoid standards were available were quantified in *Salicornia europaea* extracts using the standard addition method. The spiked extracts were analyzed in triplicate by means of targeted selection ion monitoring (SIM) scans using an isolation width of 4 amu and resolution set at 60,000.

## 4. Conclusions

This work demonstrated the potential of a nontargeted method based on Top3 ddMS^2^/MS^3^ analysis to detect and characterize flavonoids in a complex plant matrix. The selection of 15 flavonoid-specific fragment ions allowed for an efficient large-scale flagging of candidate flavonoids, including previously unreported structures. The proposed design of an advanced data processing workflow allowed for structural elucidation of unknown compounds by the establishment of structural relationships with well-known structures of flavonoids reported in the literature. The calculation of FISh scores was a relevant tool for the distinction of isomeric forms, complementary to the use of targeted MS/MS fragmentation or NMR analysis. The development of such a data processing workflow has allowed for an exploratory study of the structural diversity of flavonoids and phenolic acids and offers new perspectives in compound discovery in Salicornia extracts.

## Figures and Tables

**Figure 1 molecules-28-03022-f001:**
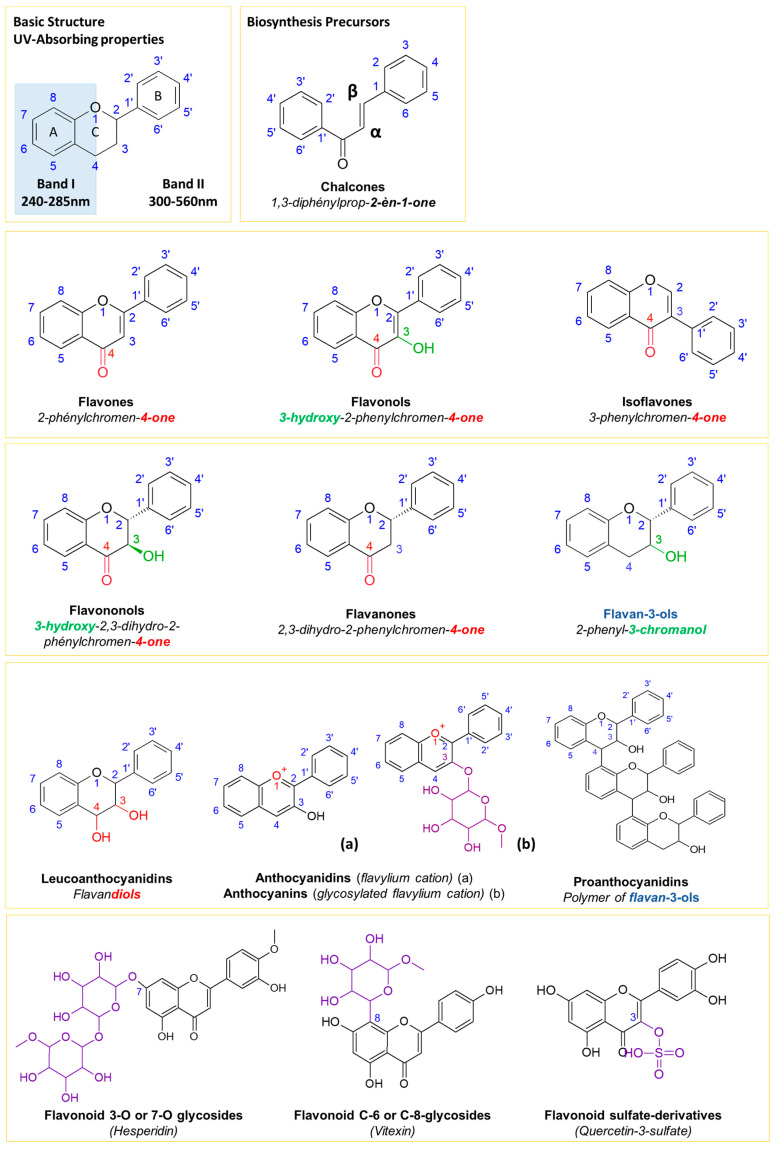
Classification of flavonoids [8,22,23,24,25].

**Figure 2 molecules-28-03022-f002:**
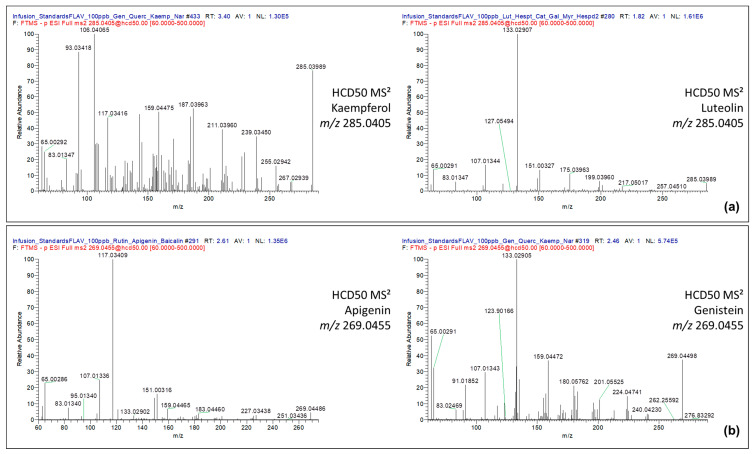
Fragmentation data acquired in HCD50 MS^2^ for the isomeric couple kaempferol/luteolin and apigenin/genistein. (**a**) Kaempferol *m*/*z* 285.0405 and Lutelin *m*/*z* 285.0405; (**b**) Apigenin *m*/*z* 269.0455 and Genistein *m*/*z* 269.0455.

**Figure 3 molecules-28-03022-f003:**
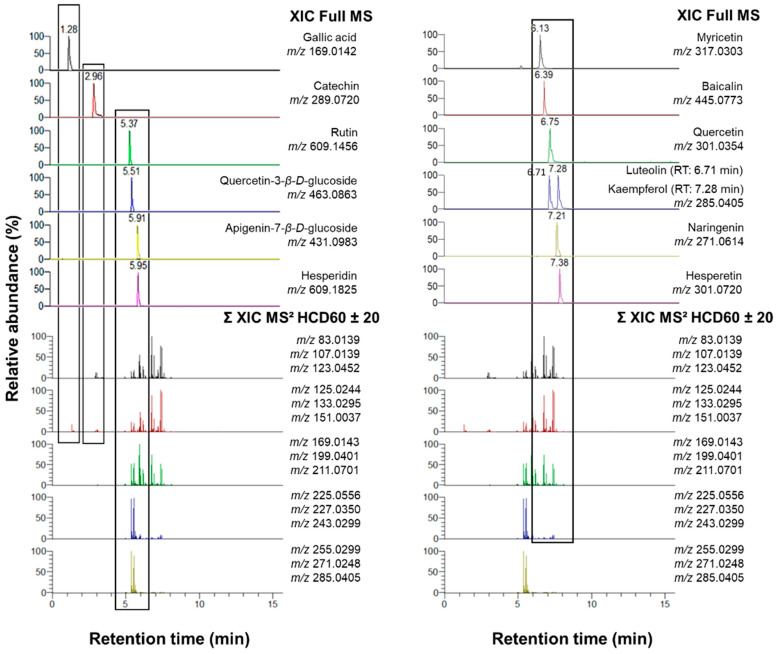
Coverage of flavonoid standards separated in reversed-phase chromatographic conditions (XIC Full MS) by the sum of extracted ion chromatogram (XIC) traces acquired in stepped HCD60 ± 20 MS^2^ scans of the 15 class-specific fragment ions.

**Figure 4 molecules-28-03022-f004:**
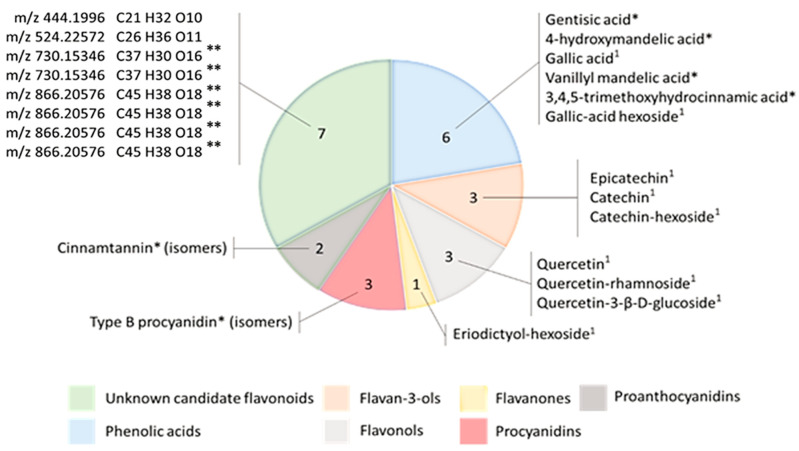
Coverage of the flavonoid contents in the model grape seed powder. Flavonoids ^1^ detected in targeted HCD30 MS^2^ analysis. Flavonoids * detected with multiple matches with online and local databases. Flavonoids ** showing several isomeric forms detected at different retention times.

**Figure 5 molecules-28-03022-f005:**
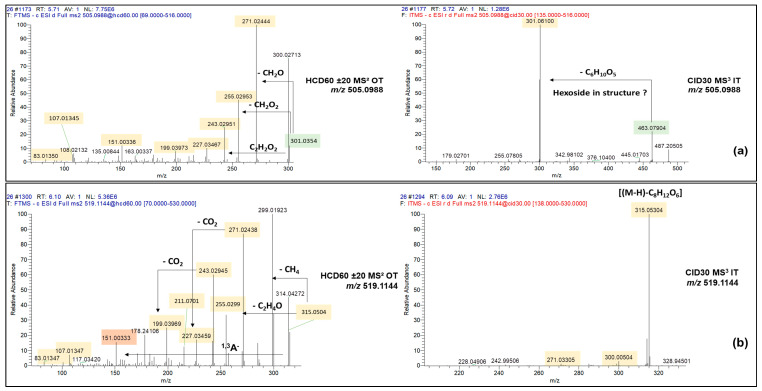
Fragmentation data acquired in HCD60 ± 20 and CID30 MS^2^ modes for the unknown candidate flavonoids *m*/*z* 505.0988 (**a**), *m*/*z* 519.1144 (**b**) and *m*/*z* 364.9973 (**c**).

**Figure 6 molecules-28-03022-f006:**
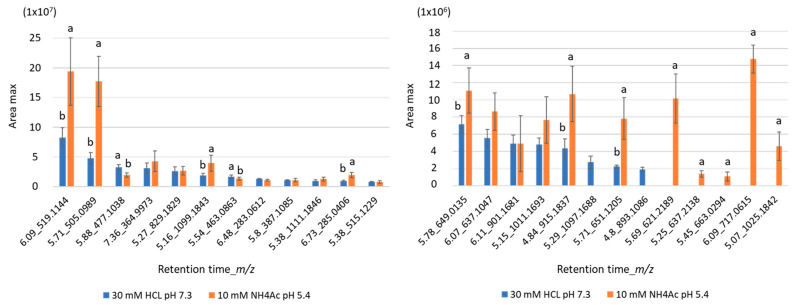
Evaluation of the extraction efficiency of flavonoids (retention time_*m*/*z*) contained in *Salicornia europaea* after ultrasound-assisted extraction in NH_4_Ac (30 mM Tris-HCl pH 7 or 10 mM NH_4_Ac pH 5.4). Area peaks are shown as mean ± SD (*n* = 3). Lowercase letters indicate statistically significant differences between area peaks observed in both extracts (*t*-test, *p* < 0.05).

**Figure 7 molecules-28-03022-f007:**
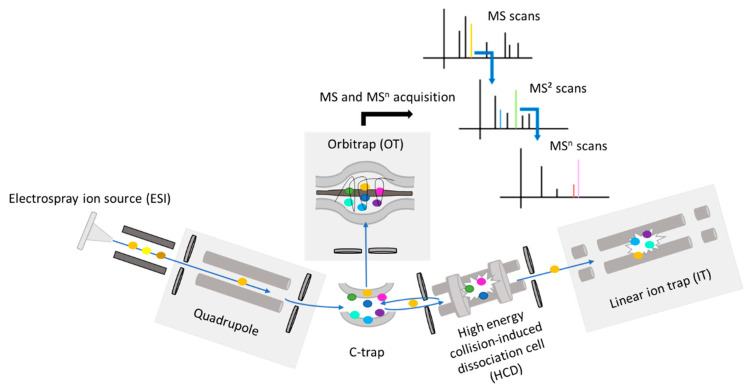
Schematic presentation for Orbitrap Fusion Lumos Tribrid^TM^ mass spectrometer. For data-dependent acquisition, precursor ions were selected in a given mass range by the use of quadrupole as scan filter. Precursors were trapped in the C-trap to be transferred to the Orbitrap for Full MS acquisition. Precursors can also be driven to the high-energy collision dissociation (HCD) cell to be fragmented under drastic voltage conditions or to the linear ion trap (LIT) to perform collision-induced dissociation (CID) fragmentation conditions. Fragment ions are in turn stored to the C-trap and sent to the Orbitrap for their detection. Note that the configuration of such a mass spectrometer permits the achievement of fragmentation levels because of its capability to store fragment ions and simultaneously perform several MS^n^ events.

**Figure 8 molecules-28-03022-f008:**
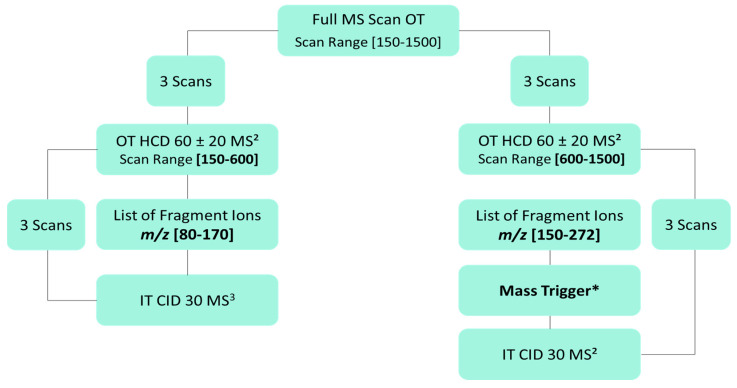
Top3 ddMS^2^/MS^3^ decision tree, driven for the nontargeted analysis of flavonoids. Two fragment ion lists were defined based on the targeted fragmentation of fifteen infused flavonoids. CID30 MS^2^ or MS^3^ scans were triggered for detected compounds showing a flavonoid-specific fragment ion in HCD60 ± 20. Mass Trigger * option permits to apply CID30 MS^2^ fragmentation on precursor to monitor neutral losses on glycosylated or sulfated flavonoid structures.

**Figure 9 molecules-28-03022-f009:**
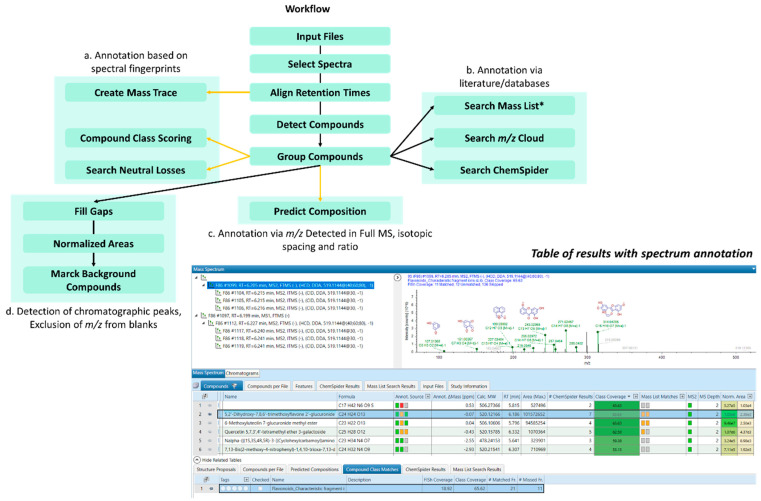
Flowchart for the untargeted workflow for the flavonoid annotation. The latter regroups different parameters, allowing for flavonoid profiling based on either their fragmentation patterns (a) or literature and databases (b). Formula was also given based on isotopic spacing and ratio (c). *Search Mass List** includes all flavonoids with their exact masses, formula and structures provided from an in-house-created database containing 85 flavonoids, and the software included *Arita Lab 6549 Flavonoid Structure Database*. *Normalized areas* and *Fill Gaps* were used for the normalization of area chromatographic peaks and comparison of their retention times in different files for enhancing compound assignments in datasets. *Mark Background* was applied for exclusion of *m*/*z* values from the blank and the reduction in signal noise.

**Figure 10 molecules-28-03022-f010:**
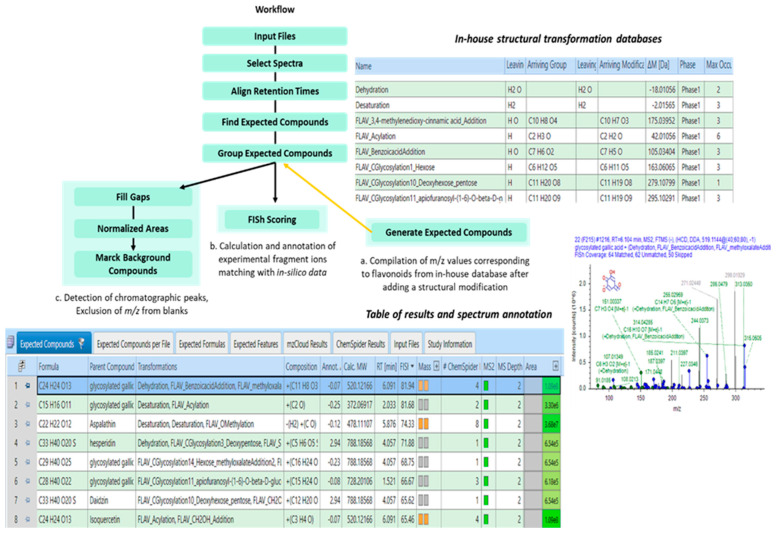
Flowchart of the data processing for the targeted screening of flavonoids. Targeted workflow designed on Compound Discoverer^TM^ to detect flavonoids in the model algae matching with those listed in the *in*-*house* database. Description of the roles of the following parameters: *Find Expected Compounds—*research of compounds in the compound list provided by the *Generate Expected Compounds* node; *Generate Expected Compounds—*creation of an exhaustive list of formulas obtained by combining all possible transformations with flavonoid structures in the in-house database;. *Group Expected Compounds—*combination of chromatographic peaks on the basis of their chemical formula and retention times; *FISh Scoring—*calculation of a score for compounds detected by the Find Expected Compounds node and annotation of the fragmentation spectra for these compounds.

**Table 1 molecules-28-03022-t001:** Flavonoid-specific fragment ions identified by the infusion of gallic acid and 14 flavonoid standards and their targeted MS^2^ fragmentation in conditions. (A) 301.0354: quercetin; (B) 269.0455: apigenin; (C) baicalein; (D) hesperidin. Fragmentation nomenclature was given based on Gates and Lopez (2012) [37], Barnaba et al. (2018) [38] and Cerrato et al. (2020) [35].

		C_4_H_3_O_2_		C_6_H_3_O_2_	C_7_H_7_O_2_		C_6_H_5_O_3_		C_8_H_5_O_3_	C_8_H_5_O_2_		C_7_H_3_O_4_		C_7_H_5_O_5_	
Class	Standards	*m*/*z* 83.0139		*m*/*z* 107.0139	*m*/*z* 123.0452		*m*/*z* 125.0244		*m*/*z* 133.0295	*m*/*z* 151.0037		*m*/*z* 169.0143		*m*/*z* 199.0401	
Phenolic acid	Gallic acid	[(M-H)-C_3_H_2_O_3_]^−^		-	-		[(M-H)-CO_2_]^−^		-	-		[M-H]^−^		-	
Flavonol	Quercetin	^1,3^A^−^C_3_O_2_		^1,3^A^−^CO_2_	-		^1,4^A^−^		^1,3^A^−^H_2_O	^1,3^A^−^		[(M-H)-C_8_H_4_O_2_]^−^		-	
Flavonol glycoside	Quercetin-β-d-glucoside	^1,3^A^−^C_3_O_2_		^1,3^A^−^CO_2_	-		-		-	^1,3^A^−^		-		[301.0354^A^-C_2_H_2_O_2−_CO_2_]^−^	
Flavonol	Myricetin	^1,3^A^−^C_3_O_2_		^1,3^A^−^CO_2_	-		^1,4^A^−^		^1,3^A^−^H_2_O	^1,3^A^−^		[(M-H)-C_8_H_4_O_3_]^−^		-	
Flavonol	Rutin	^1,3^A^−^C_3_O_2_		^1,3^A^−^CO_2_	-		-		^1,3^A^−^H_2_O	^1,3^A^−^		[301.0354^A^-C_8_H_4_O_3_]^−^		[301.0354^A^-C_2_H_2_O_2−_CO_2_]^−^	
Flavonol	Kaempferol	^1,3^A^−^C_3_O_2_		^1,3^A^−^CO_2_	^1,2^B^−^		-		-	-		-		-	
Flavanol	Catechin	^1,4^A^−^C_2_H_2_O		-	^1,3^B^−^CO		^1,4^A^−^		-	^−^		-		-	
Flavone	Apigenin	^1,3^A^−^C_3_O_2_		^1,3^A^−^CO_2_	-		-		-	^1,3^A^−^		[(M-H)-C_8_H_4_]^−^		-	
Flavone glycoside	Apigenin-7-β-d-glucoside	^1,3^A^−^C_3_O_2_		^1,3^A^−^CO_2_	-		-		-	^1,3^A^−^		[(M-H)-C_8_H_4_]^−^		[269.0455^C^-C_2_H_2_O-CO]^−^	
Flavone	Baicalin	-		-	-		-		-	-		-		[269.0455^D^-C_2_H_2_]^−^	
Flavone	Luteolin	^1,3^A^−^C_3_O_2_		^1,3^A^−^CO_2_	-		-		^1,3^B^−^	^1,3^A^−^		-		[(M-H)-C_2_H_2_O-CO_2_]^−^	
Flavanone	Hesperetin	^1,3^A^−^C_3_O_2_		^1,3^A^−^CO_2_	-		-		^1,3^A^−^H_2_O	^1,3^A^−^		-		[(M-H)-C_3_H_6_O_3_]^−^	
Flavanone	Hesperidin	^1,3^A^−^C_3_O_2_		^1,3^A^−^CO_2_	-		^1,4^A^−^		-	^1,3^A^−^		-		[301.0720^D^-C_4_H_6_O_3_]^−^	
Isoflavone	Naringenin	^1,3^A^−^C_3_O_2_		^1,3^A^−^CO_2_	-		[(M-H)-CO_2_]^−^		^1,3^A^−^H_2_O	^1,3^A^−^		-		[(M-H)-C_2_H_4_-CO_2_]^−^	
O-methylated isoflavone	Genistein	^1,3^A^−^C_3_O_2_		^1,3^A^−^CO_2_	-		-		[(M-H)-C_7_H_4_O_3_]^−^	^1,3^A^−^		[(M-H)-C_8_H_4_]^−^		-	
		**C_13_H_7_O_3_**	**C_14_H_9_O_3_**			**C_13_H_7_O_4_**		**C_13_H_7_O_5_**			**C_14_H_7_O_5_**		**C_14_H7O_6_**		**C_15_H_9_O_6_**
**Class**	**Standards**	***m*/*z* 211.0401**	***m*/*z* 225.0556**			***m*/*z* 227.0350**		***m*/*z* 243.0299**			***m*/*z* 255.0299**		***m*/*z* 271.0248**		***m*/*z* 285.0405**
Phenolic acid	Gallic acid	-	-			-		-			-		-		-
Flavonol	Quercetin	[(M-H)-CH_2_O_2_-CO_2_]^−^	-			[(M-H)-C_2_H_2_O_3_]^−^		[(M-H)-C_2_H_2_O_2_]^−^			[(M-H)-CH_2_O_2_]^−^		[(M-H)-CH_2_O]^−^		-
Flavonol glycoside	Quercetin-β-d-glucoside	[301.0354^A^-CH_2_O_2_-CO_2_]^−^	[301.0354^A^-C_2_H_2_O_2_-H_2_O]^−^			[301.0354^A^-C_2_H_2_O_3_]^−^		[301.0354^A^-C_2_H_2_O_2_]^−^			[301.0354^A^-CH_2_O_2_]^−^		[M-C_6_H_10_O_5_-CH_2_O]^−^ or [301.0354^A^-CH_2_O]^−^		-
Flavonol	Myricetin	-	-			[(M-H)-C_2_H_2_O_4_]		-			-		[(M-H)-CH_2_O_2_]^−^		-
Flavonol	Rutin	[301.0354^A^-CH_2_O_2_-CO_2_]^−^	-			[301.0354^A^-C_2_H_2_O_3_]^−^		[301.0354^A^-C_2_H_2_O_2_]^−^			[301.0354^A^-CH_2_O_2_]^−^		[M-C_12_H_20_O_9_-CH_2_O]^−^ or [301.0354^A^-CH_2_O]^−^		-
Flavonol	Kaempferol	[(M-H)-CH_2_O-CO_2_]^−^	-			-		[(M-H)-C_2_H_2_O]^−^			[(M-H)-CH_2_O]^−^		-		[M-H]^−^
Flavanol	Catechin	-	-			-		-			-		-		-
Flavone	Apigenin	-	[(M-H)-CO_2_]^−^			[(M-H)-C_2_H_2_O]^−^		-			-		-		-
Flavone glycoside	Apigenin-7-β-d-glucoside	[M-C_6_H_10_O_5_-C_2_H_2_O_2_]^−^	-			[M-C_6_H_10_O_5_- C_2_H_2_O]^−^ [269.0455^B^- C_2_H_2_O]^−^		-			-		-		-
Flavone	Baicalin	-	[M-C_6_H_8_O_6_- CO_2_]^−^ or [269.0455^C^-CO_2_]^−^			-		-			-		-		-
Flavone	Luteolin	-	-			-		[(M-H)-CH_2_O_2_]^−^			-		-		[M-H]^−^
Flavanone	Hesperetin	-	-			[(M-H)-C_3_H_6_O_2_]^−^		-			-		-		[(M-H)-CH_4_]^−^
Flavanone	Hesperidin	-	-			[M-C_12_H_20_O_9_-C_3_H_6_O_2_]^−^ or [301.0720^D^-C_3_H_6_O_2_]^−^		-			-		-		-
Isoflavone	Naringenin	-	[(M-H)-CH_2_O_2_]^−^			[(M-H)-CO_2_]^−^		[(M-H)-C_2_H_4_]^−^							
O-methylated isoflavone	Genistein	-	[(M-H)-CO_2_]^−^			[(M-H)-C_2_H_2_O]^−^		-			-		-		-

**Table 2 molecules-28-03022-t002:** Detected candidate flavonoids in *Salicornia europaea* extracts using the fragment ion search (FISh). Flavonoids for which the exact mass matched with several isomers were assigned with multiple matches *. Area peaks are shown as mean ± SD (*n* = 3).

Flavonoids	Formula	Monoisotopic Mass (Da)	Mass Error (ppm)	[M-H]^−^ *m*/*z*	Area Peak (10^6^)		RT (min)	Nb of CFI (/15)
					30 mM Tris-HCl pH 7	10 mM NH_4_Ac pH 5.4		
Multiple matches *	C_16_H_12_O_5_	284.0685	0.08	283.0612	12.6 ± 0.7	10.5 ± 1.2	6.48	10
Luteolin	C_15_H_10_O_6_	286.0478	0.33	285.0406	9.18 ± 0.8	19.4 ± 3.3	6.73	13
Luteolin-*O*-sulfate	C_15_H_10_O_9_S	366.0045	−0.06	364.9973	31.1 ±7	42.7 ± 13.3	7.36	6
Multiple matches *	C_20_H_20_O_8_	388.1158	−0.05	387.1085	10.4 ±0.9	10.5 ± 2	5.80	10
Quercetin-3-β-d-glucoside	C_21_H_20_O_12_	464.0956	0.19	463.0863	16.3 ± 2	13.2 ± 1.7	5.54	5
Isorhamnetin-hexoside	C_22_H_22_O_12_	478.1111	−0.12	477.1038	32.2 ±3	19.6 ± 2.7	5.88	9
Unknown	C_23_H_22_O_13_	506.1062	0.28	505.0989	47.8 ± 0.8	177 ± 2.7	5.71	8
Multiple matches *	C_25_H_24_O_12_	516.1301	−0.13	515.1229	7.51 ± 0.8	7.95 ± 1.7	5.38	5
Unknown	C_24_H_24_O_13_	520.1217	−0.07	519.1144	82.7 ± 12	194 ± 42	6.09	9
Quercetin-3-(6″-malonyl) -glucoside	C_24_H_22_O_15_	550.0960	0.17	549.0887	Nd	10.1 ± 2.1	5.69	9
Multiple matches *	C_30_H_38_O_14_	622.2262	0.09	621.2189	5.83 ± 0.9	8.64 ± 1.7	6.07	7
Luteolin di-hexoside	C_27_H_26_O_18_	638.1120	0.16	637.1047	Nd	1.37 ± 0.3	5.25	6
4′-OH-5,7,2′-trimethoxyflavanone 4′-rhamnoside	C_30_H_38_O_15_	638.2210	−0.08	637.2138	7.16 ± 0.7	11.7 ± 1.9	5.78	9
Unknown	C_22_H_18_O_23_	650.0211	3.37	649.0135	2.22 ± 2	7.81 ± 1.8	5.71	7
Unknown	C_28_H_28_O_18_	652.1276	5.45	651.1205	Nd	1.06 ± 0.4	5.45	9
Unknown	C_27_H_20_O_18_S	664.0368	−0.33	663.0294	Nd	14.8 ± 1.2	6.09	10
Unknown	C_27_H_26_O_21_S	718.0689	0.21	717.0615	26.7 ± 5	26.6 ± 5.7	5.27	13
Unknown	C_31_H_42_O_26_	830.1964	−0.01	829.1892	1.89 ± 0.2	Nd	4.8	4
Unknown	C_33_H_34_O_29_	894.1162	−2.49	893.1086	4.9 ± 0.7	4.91 ± 2.5	6.11	9
Unknown	C_40_H_38_O_24_	902.1757	0.48	901.1681	4.32 ± 0.9	10.7 ± 2.5	4.84	5
Unknown	C_41_H_40_O_24_	916.1911	0.11	915.1837	4.79 ± 0.6	7.66 ± 2.1	5.15	4
Unknown	C_45_H_40_O_27_	1012.175	−0.32	1011.1693	Nd	4.61 ± 1.3	5.07	10
Unknown	C_46_H_42_O_27_	1026.1914	0.13	1025.1842	2.72 ± 0.5	Nd	5.29	5
Unknown	C_48_H_42_O_30_	1098.1757	−0.32	1097.1688	18.6 ± 2.6	39.4 ± 10.2	5.16	8
Unknown	C_49_H_44_O_30_	1112.1916	−0.12	1111.1846	10 ± 1.8	12.6 ± 2.4	5.38	6

**Table 3 molecules-28-03022-t003:** Structural prediction of unknown flavonoids by means of a structural transformation database-driven data processing method.

Formula	Monoisotopic Mass (Da)	Parent Compound	Transformations	Structural Modifications	Mass Error (ppm)	FISh Coverage (%)
C_16_H_12_O_5_	284.0685	Genistein	CH_2_OH addition	+(CH_2_)	0.08	43.82
C_16_H_12_O_6_	300.0634	Naringenin	Desaturation, O-methylation	+ (CO)	0.03	30.32
C_20_H_20_O_8_	388.1158	Hesperetin	Acylation, CH_2_OH addition, O-methylation	+(C_4_H_6_O_2_)	−0.05	42.81
C_23_H_22_O_13_	506.1063	Quercetin-3-β-d-glucoside	Acylation	+(C_2_H_2_O)	0.4	42.11
C_24_H_24_O_13_	520.1217	Luteolin-*O*-glucoside	Acylation, CH_2_OH addition	+(C_3_H_4_O)	0.17	67.88
C_28_H_28_O_18_	652.1278	Glycosylated gallic acid	CH_2_OH addition, gallic acid addition	+(C_15_H_12_O_8_)	0.3	51.85
C_27_H_26_O_21_S	718.0689	6-methoxyluteolin 7-glucuronide methyl ester	Hydroxylation, methyloxalate addition, sulfation	+(C_4_H_4_O_8_S)	0.27	57.41

**Table 4 molecules-28-03022-t004:** Example of two hypothetical structures of candidate flavonoids detected in Salicornia plant.

Name	Isorhamnetin-7-*O*-(6″acetyl glucoside)	6-Methoxy-luteolin-sulfate-*O*-(dimethyloxalate-glucoside)
Structure	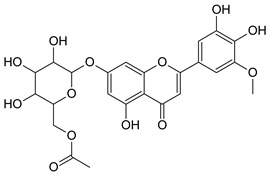	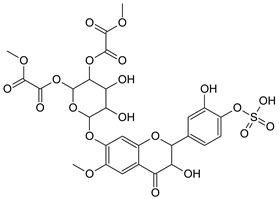
Monoisotopic mass (Da)	520.1217	718.0689
FISh score (%)	67.88	57.41

## Data Availability

All the data required for the comprehension of the present article can be found in the Appendix A.

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
