# Peer review of "Nontargeted Screening for Flavonoids in Salicornia Plant by Reversed-Phase Liquid Chromatography–Electrospray Orbitrap Data-Dependent MS2/MS3"

_molecules, 2023, doi:10.3390/molecules28073022_

Round 1

Reviewer 1 Report

Dear Authors 

Please find my comments below

1) Does matrix suppression cause any problems picking up the intense peak for your analysis? 

2) How does the infusion method HCD fragmentation pattern compare with LC-MS workflow? 

Thanks

Author Response

Dear Reviewer,
Please find in attached file the update of the manuscript "Non-targeted Screening for Flavonoids in Salicornia plant by Reversed-Phase Liquid Chromatography—Electrospray Orbitrap data-dependent MS²/MS3"

Reviewer 2 Report

The introduction is good enough to present the urgency of the research

The method good was up-to-date, reliable, and reproducible.

The narration, figures, and table have well presented the results.

The discussion part needs to improve by previous research results and updated references.

More than 50% of the references used in this article were under 2017 and pleased improve by using the updated references.

The similarity index was good enough to 14% by Turnitin.

Author Response

Dear Reviewer,

Enclosed please find the electronic file of the updated manuscript:

Non-targeted Screening for Flavonoids in Salicornia plant by Reversed-Phase Liquid Chromatography - Electrospray Orbitrap data-dependent MS²/MS3, by Maroussia Parailloux, Simon Godin and Ryszard Lobinski for consideration for publication in Molecules as a research article.

Reviewer 3 Report

In the article of Parailloux et al. entitled „ Non-targeted Screening for Flavonoids in Salicornia plant by Reversed-Phase Liquid Chromatography— Electrospray Orbitrap data-dependent MS²/MS3” Authors developed a qualitative method of characterization of flavonoids based on their fragmentation pattern from MS2/MS3 spectra.

Method validation lacks information about the number of replicates, obtained precision, accuracy, sensitivity and selectivity. All these data must be completed in order to be able to fully assess the usefulness of the proposed method.

Author Response

(The authors gave the same response as above.)

Round 2

Reviewer 3 Report

The Authors respond to the allegations. The publication can be accepted in its current form.